# Comparison of Metabolites and Gut Microbes between Patients with Ulcerative Colitis and Healthy Individuals for an Integrative Medicine Approach to Ulcerative Colitis—A Pilot Observational Clinical Study (STROBE Compliant)

**DOI:** 10.3390/diagnostics12081969

**Published:** 2022-08-15

**Authors:** Cheol-Hyun Kim, Young-Ung Lee, Kwang-Ho Kim, Sunny Kang, Geon-Hui Kang, Hongmin Chu, Sangkwan Lee

**Affiliations:** 1Department of Internal Medicine and Neuroscience, College of Korean Medicine, Wonkwang University, Iksan 54538, Korea; 2Stroke Korean Medicine Research Center, Wonkwang University, Iksan 54538, Korea; 3Hanbang Cardio-Renal Syndrome Research Center, College of Oriental Medicine, Wonkwang University, Iksan 54538, Korea

**Keywords:** gut microbes, metabolites, ulcerative colitis, integrative medicine

## Abstract

Ulcerative colitis (UC) is an intractable disease associated with high morbidity and healthcare costs. Metabolites and gut microbes are areas of interest for mainstream and complementary and alternative medicine. We, therefore, aimed to contribute to the discovery of an integrative medicine for UC by comparing and analyzing gut microbes and metabolites in patients with UC and in healthy individuals. This was an observational case-control study. Blood and stool samples were collected from the participants, and metabolite and gut microbial studies were performed. Among metabolites, formate, glycolate, trimethylamine, valine, and pyruvate levels were significantly different between the two groups. Among gut microbes, the abundance of *Bacteroidetes* at the phylum level; *Bacteroidia* at the class level; *Bacteroidales* and *Actinomycetales* at the order level; *Prevotellaceae*, *Acidaminococcaceae*, and *Leptotrichiaceae* at the family level; and *Prevotella*, *Roseburia*, *Paraprevotella*, *Phascolarctobacterium*, *Ruminococcus*, *Coprococcus*, *Clostridium_XIVB*, *Atopobium*, and *Leptotrichia* at the genus level was also significantly different. Most of the metabolites and gut microbes significantly different between the two groups were involved in energy metabolism and inflammatory processes, respectively. The results of this study could be helpful for the identification of targets for integrative medicine approaches for UC.

## 1. Introduction

Ulcerative colitis (UC) is a chronic inflammatory bowel disease (IBD) characterized by inflammation localized in the mucosa or submucosal layer of the colon. The exact cause of this condition is unknown; however, it is thought to be caused by a complex interaction of elements, including the immune system, host genotype, and environment, especially the enteric commensal microbiota [1].

UC is an intractable disease that leads to high morbidity and healthcare costs [1]. Interest in the treatment of this intractable disease in both personalized and integrative medicine has been growing [2,3]. Genes are often mentioned in personalized medicine [4]. However, although genes can be useful for predicting disease potential, treatment targeted at changing the inherited gene itself is difficult for several reasons [5]. Therefore, gut microbes and metabolites may be better targets for personalized medicine. One study reported that even if the genes are the same, disease expression may differ depending on the gut microbes [6]. In addition, for integrative medicine, there must be a common field of communication among the various medical approaches, such as Western medicine and complementary and alternative medicine. Metabolites and gut microbes could be good candidates because various medical approaches have focused on them [7,8,9].

Clinical evidence suggests that metabolites and gut microbes play a role in the pathogenesis of IBD, including UC. For example, Lavelle et al. reported that metabolites derived from gut microbes are key actors in IBD [10], and Zitomerskty et al. reported that patients with IBD who undergo surgical diversion of the fecal stream recover their uninflamed healthy intestines, but the inflammation recurs when re-exposed to the microbial laden fecal stream [11]. Another study reported that antibiotics targeting anaerobic gut microbes have shown efficacy in treating IBD [12].

Whether metabolites and gut microbes cause or result from IBD remains controversial. However, considering the existing studies, it is clear that the regulation of gut microbes and metabolites could be a new target for integrative medicine treatment [13].

The purpose of this study was to attempt to discover new targets for personalized and integrative medicine approaches to UC by comparing and analyzing gut microbes and metabolites in patients with UC and healthy individuals. Although the number of participants was small, we report our findings because we obtained significant results.

## 2. Materials and Methods

### 2.1. Study Design

This was an observational study with a case-control design.

### 2.2. Subjects

#### 2.2.1. Sample Size Calculation

As this was a pilot study and we were unable to find previous data that indicated the sample size required to produce significant findings, we relied on the recommendation made by Kieser and Wassmer that a sample size of 20–40 people be included in the pilot study [14]. From 10 December 2018 to 9 June 2020, posters in communities and hospitals were used to recruit the healthy control (HC) and UC groups. The HC group was age- and gender-matched with the UC group.

#### 2.2.2. Inclusion and Exclusion Criteria for the UC Group

The inclusion criteria for the UC group were as follows: patients diagnosed with UC who were taking UC-related drugs (e.g., anti-inflammatory drugs) agreed to participate in this study, voluntarily signed informed consent, and consumed traditional Korean dishes, such as rice and seasoned vegetables. 

The exclusion criteria were as follows: those diagnosed with diseases that could have affected the results of this study, such as diabetes mellitus and autoimmune diseases other than UC, those who had taken antibiotics or steroids within 6 months, those who were taking probiotics, those who regularly consumed alcohol and smoked, those from whom blood or stool samples could not be obtained, and those who were deemed inappropriate for participation in this study by the medical staff. 

#### 2.2.3. Inclusion and Exclusion Criteria for the HC Group

The inclusion criteria were as follows: those who consented to participate in this study freely signed informed consent, had no underlying disease, and were not taking any drugs. Those deemed inappropriate for participation in this study by the medical staff were excluded. 

### 2.3. Variables

The variables were the metabolites extracted from the collected blood samples and gut microbes extracted from the collected stool samples.

#### 2.3.1. Metabolite Analysis

##### Blood Collection Method

After 5 mL of blood was collected using the injection needle included in the blood collection kit, the blood was separated into 3.0- and 2.0-mL samples and placed in separate serum tubes and nonautologous-pooled human plasma containers, respectively. The serum and plasma were separated. 

##### Metabolite Analysis Method

A total of 250 μL of serum was combined with 500 μL of saline solution (10% D_2_O for lock signal, NaCl 0.9%, 500 mM sodium phosphate buffer in D_2_O containing 0.05 trimethylsilylpropanoic acid [TSP] 0.05% for chemical shift calibration, and concentration reference, pH 7.0). After centrifuging the samples at 12,000× *g* for 10 min, 600 μL aliquots of the supernatant were transferred to 5-mm nuclear magnetic resonance (NMR) tubes for analysis. An ASCEND 800-MHz AVANCE III HD Bruker spectrometer was used, outfitted with a 5-mm CPTIC 1H-13C/15N/DZ-GRD Z1194227/0011 cryogenic probe. The NMR sequence (Carr-Purcell-Meiboom-Gill [CPMG] condition: total T2 relaxation time of 60, 4 K data points, 128 scans, four dummy scans, 8-s delay time) used was a CPMG spin-echo pulse. The Chenomx program performed baseline correction on the 1D data obtained from the NMR analysis. Binning was then performed in units of 0.05 ppm, followed by spectral alignment using the COW algorithm in MATLAB. SIMCA −P++ was used for the multivariate analysis of the data organized using MATLAB. 

TSP was used as an internal standard for quality control. The TSP peak was used as a reference to correct for chemical shifts and quantify the metabolites. 

##### Metabolite Pattern Analysis

The signal intensity of the spectrum was normalized concerning the TSP signal and then converted into an ASCII file. An orthogonal partial least-squares discriminant analysis (OPLS-DA) was performed on the UV scale to assess differences in metabolic patterns between the HC and UC groups.

#### 2.3.2. Gut Microbe Analysis

##### Meal Adjustment Guide

The day before stool collection, participants were instructed not to drink alcohol or eat extremely fatty foods. 

##### Stool Collection and Specimen Delivery

A stool (4 mg) was placed in the stool collection kit. The outside of the kit was labeled to help distinguish specimens. The specimens were then frozen at −20 °C and transferred to the laboratory for analysis. 

##### Gut Microbe Analysis

A library was designed to enable Illumina sequencing by constructing a hybrid primer that selectively amplified the V3–V4 region of the 16S rRNA gene (the standard for identifying bacteria), and an adaptor sequence was recognized by the Illumina sequencer. According to Illumina’s MisSeq platform guide, the complete sequencing library mixture was sequenced using 300-bp paired-end sequencing. The bacteria were identified using quantitative insights into the microbial ecology pipeline after trimming the sequencing data. Greengenes was used as the bacterial identification library. A total of 20 samples that passed quality control were used in the analysis. Alpha diversity, which examines the diversity distribution of gut microbes, was compared, and a non-metric multidimensional scaling (NMDS) was performed using the Bray-Curtis distance for pattern analysis. 

#### 2.3.3. Statistical Analysis

Data collected from participants were coded and analyzed using the SPSS for Windows (version 20.0) statistical software program. The Shapiro-Wilk test was used for continuous variables to check the normality of the data. An independent *t*-test or Mann-Whitney U-test was used to compare the levels of blood metabolites and gut microbes in the stool between the UC and HC groups. To control for confounding factors, independent *t*-tests or Mann-Whitney U-tests were performed for both the sex and age groups. *p* values < 0.05 were considered statistically significant.

## 3. Results

### 3.1. Subject Characteristics

Ten patients with UC and 10 healthy individuals were recruited between 10 December 2018 and 26 February 2020. There were no significant differences in demographic characteristics, such as sex and age, between the two groups (see Table 1 for more information).

### 3.2. Metabolite Analysis

Metabolites in the UC and HC groups were clearly differentiated using principal component analysis (*R*^2^*X* = 0.563, *Q*^2^ = 0.378, Figure 1) and OPLS-DA (*R*^2^*Y* = 0.551, *Q*^2^ = 0.266, Figure 2). According to cross-validation with a 100-permutation test, the established model was considered reliable (Figure 3). Green *R^2^* values and blue *Q*^2^ values to the left were lower than the original points to the right, and the regression line of the *Q*^2^ points intersected the vertical axis below zero (*R^2^* = 0.377, *Q*^2^ = −0.157). The corresponding regression coefficients for the included metabolites sorted by their variable importance in the OPLS-DA model are shown in Figure 4. Among the metabolites analyzed, the levels of formate, glycolate, trimethylamine, valine, and pyruvate were significantly different between the two groups (*p* < 0.05). Formate, glycolate, trimethylamine, and valine levels were significantly lower, while pyruvate levels were significantly higher in the UC group than in the HC group (Figure 5).

### 3.3. Gut Microbe Analysis

The alpha diversity comparison between the two groups revealed that the UC group had significantly lower Chao1 levels, indicating lower diversity of gut microbes in this group than in the HC group (*p* = 0.013) (Figure 6). 

The NMDS based on the Bray-Curtis distance revealed that the two groups had different gut microbial patterns, but no discernable patterns were evident (Figure 7).

Significant differences in the distribution of the gut microbiota composition between the two groups were observed in *Bacteroidetes* at the phylum level; *Bacteroidia* at the class level; *Bacteroidales* and *Actinomycetales* at the order level; *Prevotellaceae*, *Acidaminococcaceae*, and *Leptotrichiaceae* at the family level; and *Prevotella*, *Roseburia*, *Paraprevotella*, *Phascolarctobacterium*, *Ruminococcus*, *Coprococcus*, *Clostridium_XIVB*, *Atopobium*, and *Leptotrichia* at the genus level (Table 2). Gut microbiota compositions at the phylum and genus levels for the UC and HC groups are shown in Figure 8 and Figure 9, respectively.

## 4. Discussion

We compared metabolites and gut microbiota between 10 patients with UC and 10 healthy individuals. The extracted metabolite mixture was analyzed via NMR spectroscopy. Afterward, a Fourier transform on the NMR data was done. The phase was adjusted to obtain a spectrum and perform baseline correction. The signal intensity of the spectrum was normalized concerning the TSP signal and then converted into an ASCII file. The converted values were analyzed using multivariate analysis. Among metabolites, univariate analysis showed formate, glycolate, trimethylamine, valine, and pyruvate levels were significantly different between the two groups. In the multivariate analysis, there were also significant differences in acetate and τ-methylhistidine between groups. Among gut microbes, the abundance of *Bacteroidetes* at the phylum level; *Bacteroidia* at the class level; *Bacteroidales* and *Actinomycetales* at the order level; *Prevotellaceae*, *Acidaminococcaceae*, and *Leptotrichiaceae* at the family level; and *Prevotella*, *Roseburia*, *Paraprevotella*, *Phascolarctobacterium*, *Ruminococcus*, *Coprococcus*, *Clostridium_XIVB*, *Atopobium*, and *Leptotrichia* at the genus level was also significantly different. The roles that these metabolites and gut microbes play are listed in Table 3. 

Van Kessel and El Aidy reported that gut microbial products are metabolites [56], and Wang et al. reported that inflammation regulates energy metabolism under physiological and pathological conditions [57]. This is consistent with the results of this study, which found that most of the metabolites and gut microbes that were significantly different between the UC and HC groups were related to energy metabolism and inflammatory processes, respectively.

Metabolites and gut microbes are areas of interest for both mainstream and complementary and alternative medicine. For example, studies have shown that herbal medicines cause metabolite change [7] and interact with gut microbes [8], and studies have shown that Western medicine also focuses on the relationship between disease and metabolites and gut microbes [9]. 

The fact that both mainstream medicine and complementary and alternative medicine are focusing on metabolites and gut microbes could have vast implications, particularly since one of the reasons that integrative medicine treatment is difficult to implement is the lack of common interests [58]. Considering these points and the results of this study, metabolites and gut microbes could be excellent targets for integrative medicine treatment.

This study had several limitations. First, because this was a pilot study, the number of participants analyzed was small. Thus, it is difficult to conclude that the results of this study reflect the characteristics of all patients with UC. However, the reliability of the results is not considered low because, despite the small number of patients, significant results were obtained that are consistent with previous research findings. Second, this study did not compare differences based on detailed information on the subjects’ diets. However, it was the same for the broad framework of traditional Korean dishes. Therefore, the possibility that diet affected the results of this study is considered insignificant. Third, this study did not evaluate the detailed correlations between the metabolites and gut microbes that showed a significant difference between the two groups. However, it was confirmed that they are commonly related to energy metabolism and inflammation. Fourth, we could not determine the names of the gut microbes that showed a significant difference between the two groups at the species level. However, we were able to confirm the lack of gut microbial diversity at the species level in the UC group through alpha diversity analysis. Fifth, it was unclear whether the patients with UC in this study were in the active or remission stage. However, it is presumed that the patients with UC included in this study were in the remission stage since those taking antibiotics and steroids, primarily used for active UC [59], were excluded. Sixth, although feces are closely related to the gut, only serum metabolites were analyzed in our study. However, considering a study by Seo [60] noted a significant difference in the metabolites in serum rather than those of the feces between chronic colitis and normal mouse models, it cannot be said that the analysis of serum metabolites in this study was incorrect.

To the best of our knowledge, most existing studies have either analyzed metabolites or gut microbes alone. However, in this study, both metabolites and gut microbes were collected from the same subjects and compared. Our data confirmed that the metabolites and gut microbes that significantly differed between the UC and HC groups were mostly related to energy metabolism and inflammation processes. If significant differences are confirmed through large-scale studies comparing metabolites and gut microbes before and after various treatments, such as with herbal medicine or Western medicine, diet, and fecal transplantation, the results could be used in developing new targets for integrative medicine approaches for UC.

## Figures and Tables

**Figure 1 diagnostics-12-01969-f001:**
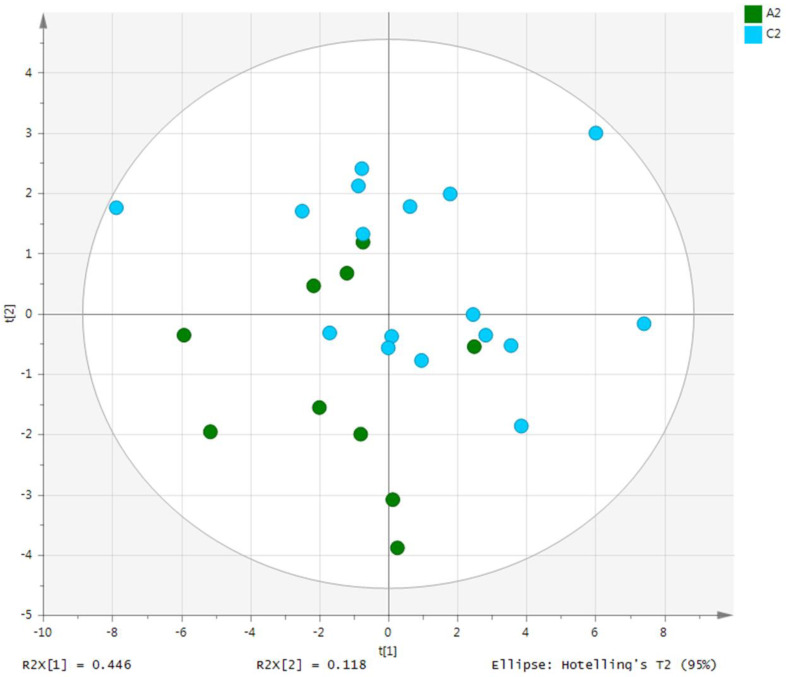
PCA score plot derived from the 1H-NMR spectra of serum from the ulcerative colitis (UC) patient group (*n* = 10) and healthy control (HC) group (*n* = 10). PCA, principal component analysis; NMR, nuclear magnetic resonance; A2, ulcerative colitis group; C2, healthy control group.

**Figure 2 diagnostics-12-01969-f002:**
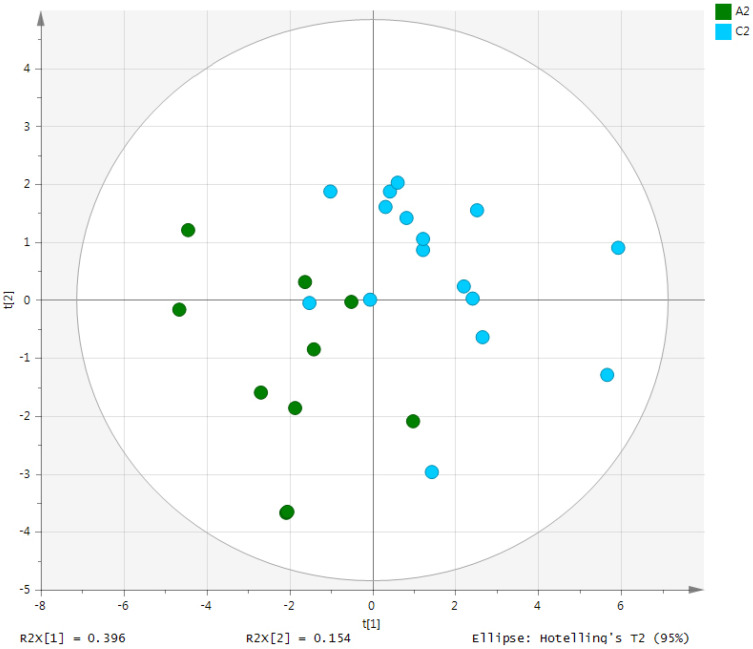
OPLS-DA score plot derived from the 1H-NMR spectra of serum from the ulcerative colitis (UC) patient group (*n* = 10) and healthy control (HC) group (*n* = 10). OPLS-DA, orthogonal partial least-squares discriminant analysis; NMR, nuclear magnetic resonance; A2, ulcerative colitis patient group; C2, healthy control group.

**Figure 3 diagnostics-12-01969-f003:**
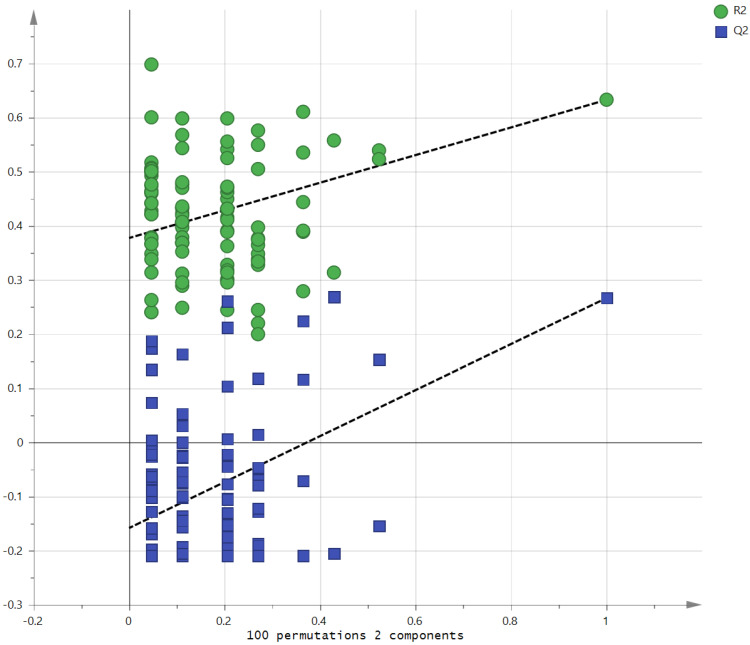
Validation of the OPLS model using the 100-permutation test.

**Figure 4 diagnostics-12-01969-f004:**
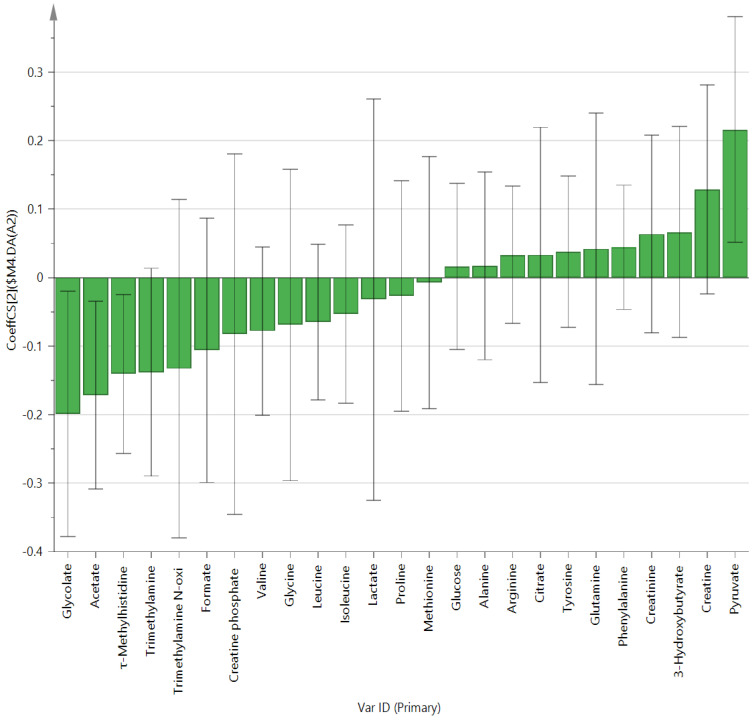
OPLS-DA coefficient plot of all metabolites in patients with ulcerative colitis.

**Figure 5 diagnostics-12-01969-f005:**
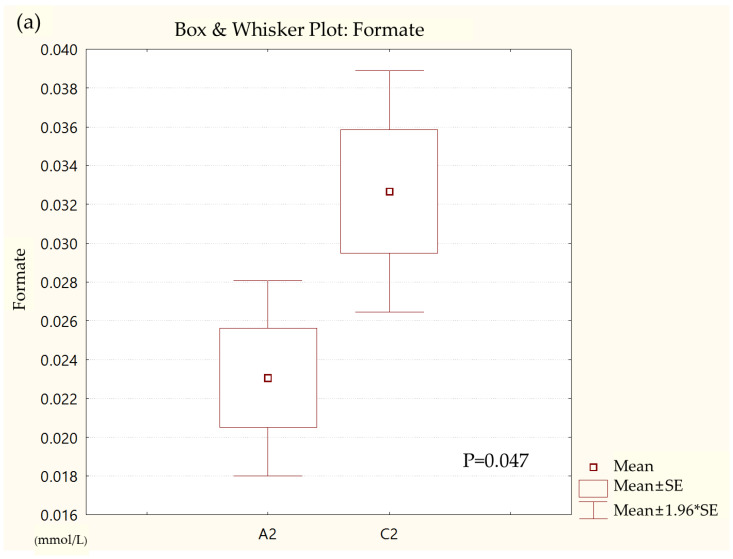
Box and whisker plot of (**a**) formate, (**b**) glycolate, (**c**) trimethylamine, (**d**) valine, and (**e**) pyruvate in ulcerative colitis (UC) patient group and healthy control (HC) group. A2, ulcerative colitis patient group; C2, healthy control group.

**Figure 6 diagnostics-12-01969-f006:**
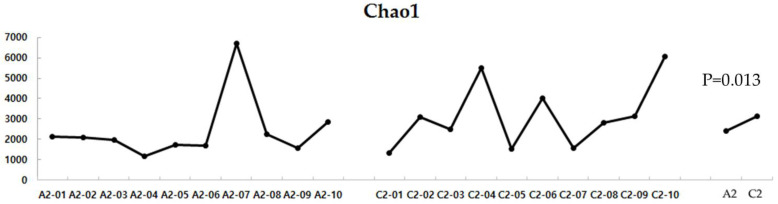
Alpha diversity of the UC and HC groups. UC, ulcerative colitis; HC, healthy control; A2, ulcerative colitis patient group; C2, healthy control group.

**Figure 7 diagnostics-12-01969-f007:**
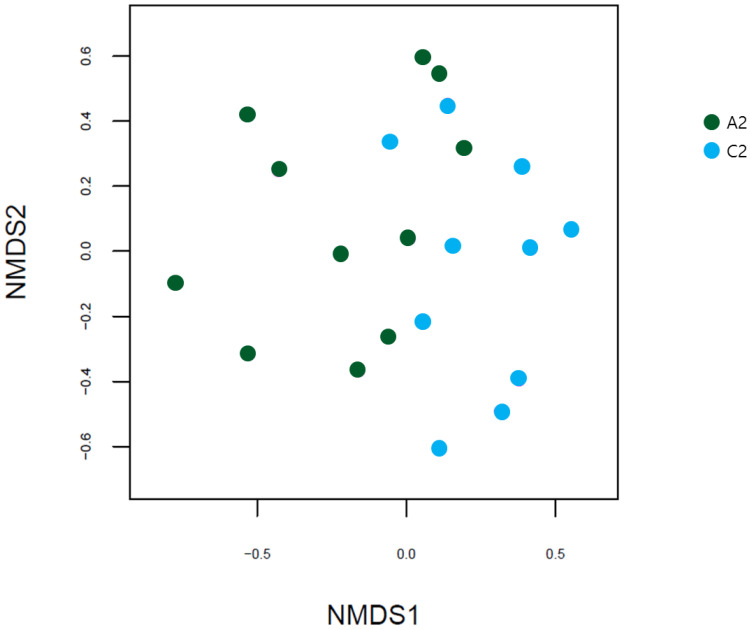
NMDS plots based on Bray-Curtis distances between the UC and HC groups. NMDS, non-metric multidimensional scaling; UC, ulcerative colitis; HC, healthy control; A2, ulcerative colitis patient group; C2, healthy control group.

**Figure 8 diagnostics-12-01969-f008:**
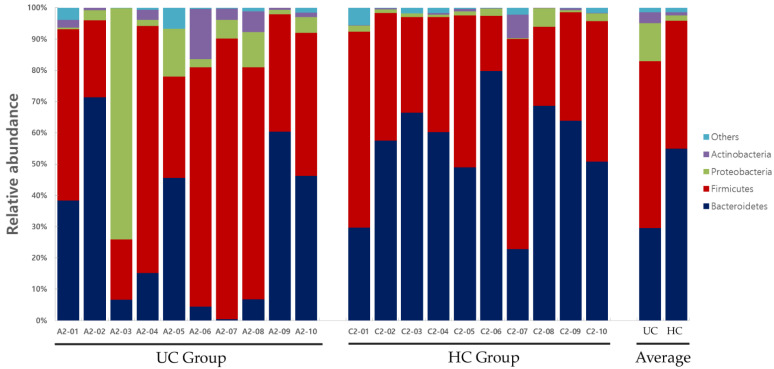
Gut microbiota composition at the phylum level in UC and HC groups. UC, ulcerative colitis; HC, healthy control.

**Figure 9 diagnostics-12-01969-f009:**
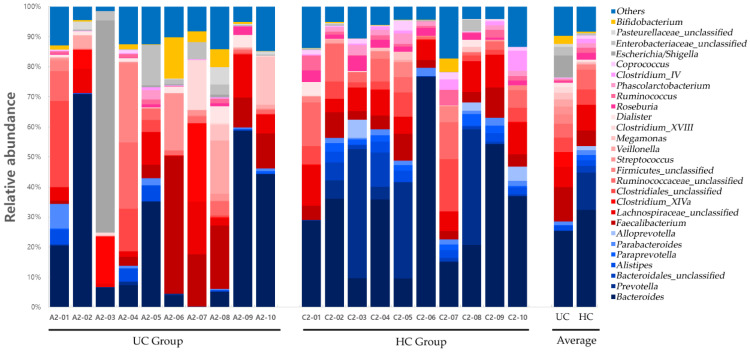
Gut microbiota composition at the genus level in UC and HC groups. UC, ulcerative colitis; HC, healthy control.

**Table 1 diagnostics-12-01969-t001:** Demographic characteristics and medical history of enrolled subjects.

Classification		UC Group	HC Group	*p* Value
Total		10	10	
Sex	Male	5	5	*p* > 0.05
	Female	5	5	
Age (years)	Minimum	33	33	*p* > 0.05
	Maximum	77	72	
	Average	59.4	53.9	
Disease duration (years)	Minimum	2	-	
	Maximum	18	-	
	Average	9.4		
Comorbidities	Hypertension	5	-	
	Dyslipidemia	1	-	
	Prostatic hypertrophy	1	-	
	None	5	10	
Active ingredients in the medications taken	Mesalazine	8	-	
	Sulfasalazine	2	-	
	Rebamipide	6	-	
	Pinaverium bromide	4	-	
	Itopride hydrochloride	3	-	
	Mosapride citrate hydrate	1	-	
	Telmisartan	1	-	
	Amlodipine besylate	2	-	
	Losartan potassium	2	-	
	Carvedilol	1	-	
	Olmesartan medoxomil	1	-	
	Atorvastatin calcium trihydrate	1	-	
	Finasteride	1		
	None	0	10	

UC, ulcerative colitis; HC, healthy control.

**Table 2 diagnostics-12-01969-t002:** Gut microbiota compositions according to taxonomic level in UC and HC groups.

Classification		Gut Microbes	UC Group vs. HC Group	
			↑/↓ ^§^	Significance (*p* < 0.05)
Stool	Phylum level	*Bacteroidetes*	↓	0.022
	Class level	*Bacteroidia*	↓	0.023
	Order level	*Bacteroidales*	↓	0.023
		*Actinomycetales*	↑	0.044
	Family level	*Prevotellaceae*	↓	0.020
		*Acidaminococcaceae*	↓	0.015
		*Leptotrichiaceae*	↑	0.025
	Genus level	*Prevotella*	↓	0.049
		*Roseburia*	↓	0.016
		*Paraprevotella*	↓	0.011
		*Phascolarctobacterium*	↓	0.016
		*Ruminococcus*	↓	0.015
		*Coprococcus*	↓	0.028
		*Clostridium_XIVB*	↓	0.049
		*Atopobium*	↓	0.015
		*Leptotrichia*	↑	0.038

UC, ulcerative colitis; HC, healthy control. § Arrows (↑ and ↓) indicate a decrease or increase in microorganism levels in patients with UC compared with healthy individuals.

**Table 3 diagnostics-12-01969-t003:** Description of metabolites and gut microbes that significantly differed between UC and HC groups.

Classification			Description
Metabolites		Formate	Formate is associated with glucose-lactate metabolism. Immunologically, it is related to the decline of naïve T cells [15]. Formate also plays a role in producing energy through anaerobic respiration as an electron donor [16].
		Glycolate	Glycolate is a major precursor to oxalate [17], which is closely related to stone disease [18], and according to a report by Caudarella et al., stone disease occurs more commonly in patients with IBD [19].
		Trimethylamine	Trimethylamine is caused by the intestinal degradation of dietary constituents such as choline and carnitine by microbial enzymes [20]. Trimethylamine is also a precursor to trimethylamine-N-oxide, which is associated with the risk of athero-thrombogenesis [20]. According to a study by Alfredo et al., IBD is closely associated with the risk of thrombotic complications [21]. Marchesi et al. also analyzed the metabolites of patients with IBD through fecal samples and found a decrease in trimethylamine, which is consistent with our study [12].
		Valine	Valine is a minor substrate of brain energy metabolism. During glutamatergic signaling, valine metabolism appears to be particularly crucial in the process of glutamate translocation between astrocytes and neurons [22]. Valine is an essential amino acid in animals, including humans, and must be ingested into the diet [23].
		Pyruvate	Pyruvate is the end-product of glycolysis. Abnormal pyruvate metabolism plays an especially prominent role in cancer, heart failure, and neurodegeneration. It is also associated with chronic obstructive pulmonary disease, obesity, diabetes, and aging [24].
		Acetate	Acetate is a short-chain fatty acid (SCFA) produced by gut microbes, which regulates inflammation in inflammatory and metabolic diseases [25]. Deleu et al. reported that SCFAs, including acetate, are closely related to IBD [26].
		τ-Methylhistidine	τ-Methylhistidine is associated with the degradation of intestinal proteins [27]. Wang et al. suggested that τ-methylhistidine is one of the potential biomarkers for ulcerative colitis [28].
Gut microbes	Phylum level	*Bacteroidetes*	*Bacteroidetes* are known to produce anti-inflammatory metabolites such as SCFAs [29]. Our research team has previously confirmed that *Bacteroidetes* levels are lower in patients with Parkinson’s disease than in healthy individuals, which is related to neuroinflammation [30].
	Class level	*Bacteroidia*	*Bacteroidia* dominate microbial communities inhabiting the anaerobic environment of the lower gastrointestinal tract. Metabolic end products generated by *Bacteroidia* change the nutritional environment for both the host and other intestinal microbes. Formate, which was significant in the results of this study, is also a metabolic end product of *Bacteroidia* [16].
	Order level	*Bacteroidales*	*Bacteroidales* have been found to modulate host immunological and intestinal activities such as mucosal barrier fortification, intestinal immune maturation, and angiogenesis by occupying a vital niche at the mucosal surface of the intestine. *Bacteroidales* species can have positive or harmful effects on their hosts, depending on their genetic content. In patients with IBD, more severe inflammation has been correlated with lower *Bacteroidales* diversity [31].
		*Actinomycetales*	Many *Actinomycetales* found in natural substrates can prevent bacteria and other microbes from growing [32]. In one study, *Actinomycetales* were higher in patients with irritable bowel syndrome than in normal subjects [33]. Based on these studies, the decreased intestinal microbial diversity in patients with IBD may be related to the abundance of *Actinomycetales*.
	Family level	*Prevotellaceae*	The *Prevotellaceae* family is associated with antibiotic biosynthesis and the transport of secondary metabolites [34]. Generally, *Prevotellaceae* produce SCFAs through the fermentation of dairy products. Reduced SCFAs cause increased gut permeability, which exposes the intestine to bacterial endotoxins [35]. In another study, the number of *Prevotellaceae* and *Prevotella* was significantly lower in patients with UC than in controls [36].
		*Acidaminococcaceae*	The family *Acidaminococcaceae* is now called *Veillonellaceae*. The *Veillonellaceae* family is implicated in regulating systemic inflammation [37] and is therefore presumed to be closely related to immune-mediated inflammatory disease, including IBD [38]. In one study, it was suggested that *Veillonellaceae* might be a gut microbe closely related to IBD [39].
		*Leptotrichiaceae*	*Leptotrichiaceae* generally inhabit mucous membranes, but when introduced into different tissue or host sites, they can shift their pathogenic potential and produce severe and even life-threatening disease, according to their phylotypes [40].
	Genus level	*Prevotella*	*Prevotella* is thought to be closely related to chronic inflammation [41], with one study reporting a reduction in *Prevotella* in pediatric patients with Crohn’s disease [42].
		*Roseburia*	*Roseburia*, one of the most common gut microbes, is decreased in patients with IBD. It helps to protect the mucosa of the colon from inflammation and subsequent IBD. Therefore, *Roseburia* could be a candidate for IBD treatment [43].
		*Paraprevotella*	The primary fermentation products of *Paraprevotella* are succinic acid and acetic acid, which are associated with inflammation. Acetic acid is especially known to alleviate inflammation [44,45].
		*Phascolarctobacterium*	*Phascolarctobacterium* is already known to be associated with IBD. These bacteria are presumed to produce propionate, which has been found to have anti-inflammatory properties [46,47].
		*Ruminococcus*	*Ruminococcus* has been associated with intestinal inflammation and is less abundant in patients with IBD [48]. *Ruminococcus* help their hosts degrade and convert complex polysaccharides into various nutrients [49].
		*Coprococcus*	The association of *Coprococcus* with IBD has long been reported. Agglutinating antibodies for *Coprococcus* were briefly considered a biomarker for IBD [50]. In autoimmune diseases, the relative abundance of *Coprococcus* is lower, and the guts of patients with an autoimmune disease have been characterized by a reduction in microbes, which is positively correlated with heptanoate and hexanoate [51]. Heptanoate and hexanoate belong to SCFAs and are involved in the inflammation process [52].
		*Clostridium_XIVB*	The genus *Clostridium*, including *Clostridium_XIVB*, plays a role in modulating the biosynthesis and release of serotonin [53]. The majority of serotonin is produced in the gastrointestinal epithelium, where it is suggested to act as a prominent regulatory molecule in the IBD [54].
		*Atopobium* *Leptotrichia*	*Atopobium* and *Leptotrichia* are oral microbes swallowed with saliva into the digestive tract. The dysbiosis of oral microbes, including *Atopobium* and *Leptotrichia,* can trigger gut microbe dysbiosis, leading to IBD [55].

UC, ulcerative colitis; HC, healthy control.

## Data Availability

The data in this study are available from the corresponding author upon reasonable request.

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
