# Peer review of "Comparison of Metabolites and Gut Microbes between Patients with Ulcerative Colitis and Healthy Individuals for an Integrative Medicine Approach to Ulcerative Colitis—A Pilot Observational Clinical Study (STROBE Compliant)"

_diagnostics, 2022, doi:10.3390/diagnostics12081969_

Round 1

Reviewer 1 Report

Overall, an excellent manuscript is submitted. The pilot study design is appropriate. All figures and tables clearly show the results. Adequate discussion and conclusions are given. English text is acceptable. To be accepted in present form.

The submitted manuscript is with high scientific value, novelty and is of great interest to readers from Biomedical research community.

Author Response

Thank you for reviewing the manuscript. Your comments and suggestions have helped us improve our manuscript.

Reviewer 2 Report

The authors present an observational case study describing differences in gut microbiota and systemic metabolome of a small number of UC patients and healthy controls. Although mostly soundly executed, from the diagnostics point of view this study does not bring any novelties. It is, therefore, my opinion that its merits for publication in Diagnostics are very low and that other type of journal would be more appropriate.

Some minor remarks:

1.       Indicate the Institution responsible for ethical approvals of the study.

2.       In metabolomics analysis, description of methods gives impression that different normalization methods were used for metabolite analysis: total spectral area for multivariate analysis and the area of TSP peak for metabolite quantification followed by univariate analysis? Also, the results of these two approaches were not discussed.

3.       Some latest references related to the use of metabolomics in UC and/or integrative medicine would be welcome.

4.       Indicate the units of y axes in Figure 5.

Compliant)

Author Response

Thank you for reviewing the manuscript. Your comments and suggestions have helped us improve our manuscript. Please see the attachment. 

Reviewer 3 Report

This clinical study was not very innovative. The authors need to address the following issues.

1. Why were the blood test results used to determine that the volunteers meet the test criteria not shown in the results?

2. Were the UC patients in this study in active or remission?

3. Feces are more closely linked to the gut, but why only serum metabolites and not fecal metabolites are tested?

4. The clarity of the figures in the article is generally low, especially Figure 5, please modify.

5. Please add the units of metabolites in Figure 5.

6. There is no corresponding figure for Chao 1 level described in line 189, please add it.

7. The PCoA analysis is described in line 192, but the NMDS analysis is shown in Figure 6, which is different, please correct it.

Author Response

(The authors gave the same response as above.)

Round 2

Reviewer 2 Report

Thank you for revising the manuscript. I would only have a few minor points.

CCould the authors also change the Methods and Material section (2.3.1.3) which still reads “The 1H NMR spectrum was normalized to the total area” according to the text added to Discussion: The signal intensity of the  spectrum  was  normalized  with  respect  to  the  TSP signal, and the intensity of the signal was converted into an ASCII file. 

Also, MVA demonstrated (based on OPLS-DA coefficient and presumably the corresponding confidence intervals as the selection criteria) that apart from pyruvate and glycolate, acetate and methylhistidine also changed. Although the authors discuss well the changes in metabolites that were significantly changed when evaluated by univariate analysis (p<0.05 for formate, glycolate, trimethylamine, valine, and pyruvate), they do not try to discuss the metabolites identified by MVA. Could you also include acetate and methylhistidine to your discussion?

Author Response

Thank you for reviewing the manuscript. Your comments and suggestions have helped us to improve our manuscript. Please see the attachment. 
